# From Counseling Transcript to Mind Map: Leveraging LLMs for Effective Summarization in Mental Health Counseling

## Abstract

The increasing number of patients with mental health illness has heightened the cognitive load on therapists, making it challenging for them to provide personalized care that each patient requires. Summarizing counseling sessions can aid mental health practitioners in recalling key details. However, most existing research on summarization focuses primarily on text-based summaries which often require significant cognitive effort to read and interpret. Visual-based summary such as mind maps is proven to help enhance cognitive understanding by giving a quick overview of topics and content. Nevertheless, due to the complex nature of counseling which involves substantial qualitative data, generating visual-based summaries using traditional AI models can be challenging. With the recent advancements in Large Language Models (LLMs), these models have demonstrated the capability to perform tasks based on instructions and generate outputs in various formats. In this study, we develop a web-based summarization tool that serves as a pipeline in performing summarization of counseling transcripts into visual-based mind map summaries using LLMs. We conducted a human evaluation to validate the effectiveness of the generated visual-based summary based on criteria of accuracy, completeness, conciseness and coherence. Our findings show that our web-based summarization tool can effectively extract key points from counseling transcripts and present them in visual-based mind maps, demonstrating its potential in enhancing insights for therapists, ultimately simplifying the process of documenting counseling sessions.

## 1 Introduction

The World Health Organization (WHO) emphasizes that there can be "no health without mental health" (WHO, 2005). This statement holds true as mental illness affects one in four individuals during their lifetime (Ginn & Horder, 2012). While common mental health conditions such as stress or mild depression can often be self-managed, a fraction of the population encounters more serious disorders, including schizophrenia and severe depression. These serious mental disorders typically require professional counseling treatment such as receiving psychotherapy (Althoff et al., 2016; Kuo et al., 2022). Generally, mental health counseling follows a well-structured approach, which may contain discussions about the patient's personal experiences and the emotional challenges that are affecting their mental state (Srivastava et al., 2022). To build a strong therapeutic relationship with the patient, therapists are required to establish trust and create a comfortable environment for open communication. One effective way to achieve this is by empathizing with the patient's perspective and gaining a deep understanding of what the patient is trying to convey (Ivey et al., 2018). With the concerning surge in the number of patients with mental health illness in recent years (Panchal et al., 2020), it has certainly become challenging for therapists to manage their caseloads effectively hindering the delivery of personalized care each patient needs (Malhotra et al., 2022; Werbart et al., 2019). The heightened cognitive load resulting from these overwhelming circumstances could be reduced by implementing automated summarization of counseling session transcripts. While most therapists are trained to take notes during therapy sessions, having an automated summarization of the session's context would enhance their ability to recall key details and gain a deeper understand-

ing of their patients. Additionally, the summarization tool could help verify therapists' memories, ensuring accuracy and consistency in their understanding of each session.

With the advancements in the field of Natural Language Processing (NLP), dialogue summarization is no longer a new concept. Several previous studies (Malhotra et al., 2022; Lee et al., 2019; Zhang et al., 2021) have explored methods for summarizing transcripts from dialogues, with some focusing specifically on extracting and summarizing dialogue acts. This can provide therapists with a clearer understanding of key interaction points, enhancing their ability to analyze client's responses. The outputs of these studies were primarily presented in text-based formats. Although text-based outputs are effective, incorporating visual elements such as mind maps can further enhance clarity and facilitate a quicker understanding of the counseling session. This is because text-based summaries generally require more cognitive effort to read and interpret which can be time consuming particularly when dealing with dense information. Figure 1 and Figure 2 shows the comparison between both text-based and visual-based summaries of a counseling session. As depicted in Figure 2, visualization techniques can present summarized information by categorizing key points and organizing them hierarchically. This approach reduces information complexity and makes the content more engaging and memorable (Hochheiser & Verma, 2021). However, developing visual-based summaries is highly subjective to the context of the counseling conversation, which primarily involves qualitative data. This makes it challenging for therapists to create such summaries in real-time dialogue environment. Nevertheless, our work demonstrates that generating these visual summaries is feasible with the capabilities of LLMs.

> The patient loves their job and has been very helpful by saving kids from dangerous sitations. But it doesn't feel like that anymore. The patient is afraid recently of making a mistake in their job. The patient does the same as before, but now they are not able to trust their own judgement. They are having self doubts and second guessing everything.

Figure 1: An example of text-based summary extracted from the MEMO dataset (Srivastava et al., 2022).

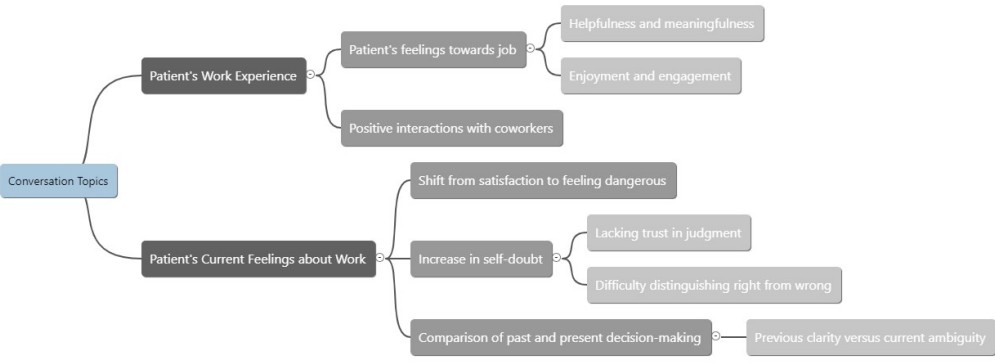

Figure 2: An example of a generated visual-based summary from our web-based summarization tool.

In this paper, we first develop a web-based summarization tool which serves as a pipeline to summarize counseling transcripts into visual-based summaries by leveraging LLMs. State-of-the-art LLMs such as GPT-4 represent a significant breakthrough in NLP. Their ability to perform various tasks, including summarization, classification, translation and creative writing has attracted many users from diverse fields, including mental health. The capabilities of LLMs in processing text could potentially revolutionize mental health practices. Our tool incorporate prompt engineering methods like one-shot prompting to facilitate the model in extracting key points from the counseling transcripts and

converting the output into mind maps. This tool fills a critical gap in existing summarization tools which generally provide textual summaries without considering how visual representations can improve understanding and engagement. Our approach enables therapists to quickly visualize key topics and content from counseling sessions thereby improving efficiency in recalling key details of the session. We employ the MEMO[1] dataset which consists of 212 counseling conversations between the therapists and patients. These counseling transcripts are sourced from publicly available material, primarily from YouTube. Each conversation is paired with a human-written text-based summary (reference summary) that is provided in the dataset. These summaries serve as a benchmark for our study. For this study, we subsample the dataset to use only 20 samples, ensuring a more focused and manageable set for our evaluation survey. A one-shot example of the desired output, along with instructions for the model to summarize the transcript is provided as a prompt to the language model we use, GPT-4o Mini. The model generates output in PlantUML mind map syntax which is then converted into an interactive mind map as shown in Figure 2. The generated output is evaluated by human reviewers. We invited several researchers from the School of Information Technology at Monash University Malaysia to participate in an evaluation survey, where they compared the content of the generated mind map with the text-based summary. The assessment was based on various criteria, including accuracy, completeness, conciseness, and coherence.

We summarize our main contributions of our study as follows:

- We developed a web-based summarization tool that uses LLMs and a diagramming tool to create visual-based summaries from counseling session transcripts.

- We leverage the capabilities of state-of-the-art LLMs specifically GPT-4o Mini to devise a prompt suitable to summarize key points from counseling transcripts and formatting the output to PlantUML mind map syntax.

- We propose utilizing mind maps as an innovative visual format for summarizing counseling sessions, shifting away from conventional text-based summaries. This visual approach enhances comprehension and reduces cognitive load for therapists by organizing key points hierarchically. Ultimately, helping to improve the efficiency of care delivery in mental health settings.

## 2 LITERATURE REVIEW

### 2.1 DIALOGUE SUMMARIZATION

Dialogue summarization is the process of condensing a conversation into a shorter version that covers the important context from the original dialogue (Feng et al., 2021). These summaries are typically presented in text form and can vary in length depending on the dialogue's context density. Generally, summarization methods are categorized into two broad types: extractive and abstractive (Srivastava et al., 2022). Extractive methods select key information from the dialogue and present it in sentences without significantly altering their original semantics. In contrast, abstractive methods generate summaries that may include words or sentences not present in the dialogue. For example, abstractive summaries involve paraphrasing the content of the conversation, typically aiming to improve the overall flow and coherence of the text (Sharma & Sharma, 2022).

Automatic summarization has been a subject of study for a long period of time (Paice, 1990). From early methods like keyword frequency analysis to modern AI-based models, the field of automatic summarization has certainly evolved over time. Several studies have emerged in the recent years, proposing AI-based strategies for producing abstractive summaries for medical and mental health conversations. For example, Zhang et al. (2021) explored the summarization of doctor-patient dialogues by fine-tuning various pretrained transformer models, with a focus on the patient's history of present illness (HPI). Similar methods can be found in other research, such as Srivastava et al. (2022), which focuses on summarizing counseling components. Their study involves an utterance classification model which acts as a guide for their summarization model to generate abstractive summaries based on each criterion.

The growing popularity of LLMs has further motivated research in this area. For example, a study by So et al. (2024) demonstrated the use of LLMs to summarize stressors and symptoms from interview

---

[1] https://github.com/LCS2-IIITD/MEMO

transcripts of ten North Korean defectors. The study primarily explores the potential of LLMs to identify and categorize utterances that reflect psychiatric symptoms in patients. The researchers further tested the LLMs in summarizing these identified symptoms and stressors. Their results showed that LLMs can successfully perform these tasks, highlighting their effectiveness in analysing complex transcripts. Furthermore, a survey by Jin et al. (2024) reported that the performance of LLMs in text summarization is comparable to human-written summaries based on evidence from previous studies focusing on text summarization (Basyal & Sanghvi, 2023; Laskar et al., 2023). However, these studies predominantly focus on text-based outputs either using abstractive or extractive methods. The development of visual-based summaries remains underexplored. This may possibly be due to the complexity of using Deep Learning methods for such outputs which require sophisticated training and testing processes. Nevertheless, the advancements of LLMs in capturing nuanced relationships within text data and the their ability to format outputs based on specific requirements open up new possibilities. In this study, we leverage LLMs to generate summaries in PlantUML mind map format, then translating the generated text-based syntax into visual-based summaries.

## 2.2 Information Visualization

Information visualization involves the use of visual representations to depict abstract non-physical data in order to enhance cognitive understanding (Card, 1999; Chan et al., 2024). This approach is widely applied across various domains including education, healthcare, finance and mental health. For instance, bar charts, line charts and pie charts can effectively display information about users' health. Studies indicate that presenting clear and meaningful data helps users quickly comprehend their health conditions and can significantly improve their motivation towards a healthier lifestyle (Chan et al., 2024).

In the mental health domain, various research efforts have led to solutions aimed at improving mental well-being. Mobile apps have been developed to monitor users' mental health status. For example, Emotion Guru, developed by van Cuylenburg & Ginige (2021) is an emotion-tracking application that detects emotions through social media posts on Facebook. The app accesses the user's Facebook posts and calculates the sentiment polarity for each post. A bar chart is used to depict the overall positive and negative posts from the past 24 hours. Additionally, there is a mood tracking feature where users can manually record their mood each day. This data is then presented in a line chart, showing mood trends over the past 7 days. A similar study by Yamashita et al. (2019) who developed a mobile app called PNViz (Positive-and-Negative Polarity Visualizer). This app captures the level of positivity through short voice messages recorded by users. The results are then displayed in a line graph allowing users to visualize trends of their mental health positivity levels.

Thus far, in the mental health field, visualization techniques have primarily been designed to help patients better understand their own mental well-being. However, little attention has been given to developing strategies that assist practitioners. Chan et al. (2024) conducted a review and found that only one of seven studies focused on visualization techniques designed specifically for practitioners. These studies mainly visualize quantitative data such as mood trends, which can be easily plotted on charts. As mental health practitioners often conduct counseling sessions, the data they work with is primarily qualitative rather than quantitative, making it more complex to represent through traditional visualization methods (Hochheiser & Verma, 2021). The subjective nature of conversations, emotions and experiences captured in these sessions requires more nuanced approaches to visualization to provide meaningful insights for both practitioners and patients.

In this study we utilize PlantUML[2], a diagramming tool that provides a simple markdown syntax for representing mind maps. This tool can facilitate LLMs to generate outputs in such formats, making it feasible to organize key points of the counseling session in a hierarchical structure.

## 3 Methodology

This section outlines the dataset used as input for the LLM along with the experiment setup and the evaluation process. Figure 3 depicts the overall methodology of this study and our web-based summarization tool.

---

[2]https://plantuml.com/mindmap-diagram

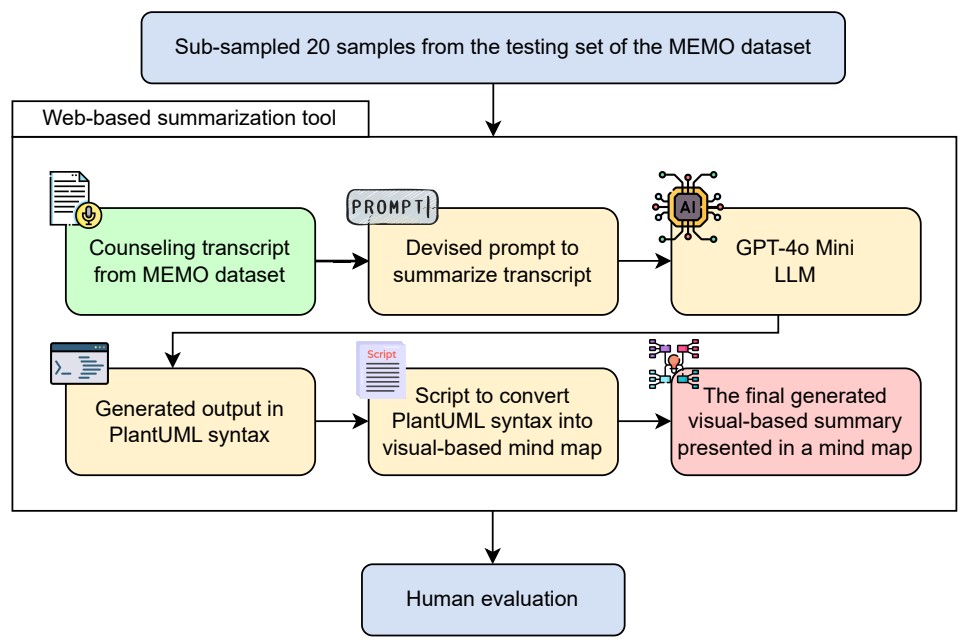

Figure 3: Overview of the methodology of this study and our web-based summarization tool.

Table 1: Overview of the MEMO dataset (Srivastava et al., 2022).

| Split | Number of Samples | Speaker | Number of Utterances |
|-------|-------------------|---------|----------------------|
| Train | 152 | Patient | 4766 |
|       |     | Therapist | 4877 |
| Test | 39 | Patient | 1004 |
|      |    | Therapist | 1006 |
| Validation | 21 | Patient | 594 |
|            |    | Therapist | 597 |
| Total | 212 | Patient | 6364 |
|       |     | Therapist | 6480 |

## 3.1 DATASET

In this study, we employ the Mental Health Summarization (MEMO) dataset as our main dataset (Srivastava et al., 2022). The MEMO dataset is an extended version of the HOPE[3] dataset which contains 12.9K utterances from 212 counseling sessions between the therapist and patient (Malhotra et al., 2022). These utterances are transcribed from counseling videos gathered from publicly available sources such as YouTube, all of which involve dyadic conversations. The transcripts have been further preprocessed to include speaker labels (P - patient, T - therapist) for each utterance and to correct transcription errors. Additionally, the transcripts are fully anonymized with actual names replaced by synthetic ones. As shown in Table 1, the dataset is split into training, testing and validation sets following a 70:20:10 ratio with 152 samples for training, 39 for testing and 21 for validation.

---

[3]https://github.com/LCS2-IIITD/SPARTA_WSDM2022

The study by (Srivastava et al., 2022) focuses on counseling summarization guided by the identification of key counseling components. To support their study, the authors conducted a comprehensive annotation process which was guided and validated by mental health experts. Each utterance was labeled with relevant counseling components, specifically: *symptom and history*, *patient discovery*, *reflecting* and *discussion filler*. Additionally, a reference summary was created for each counseling conversation. This reference summary serves as a baseline text-based summary in our study allowing us to compare and evaluate the generated visual-based summary against it.

## 3.2 EXPERIMENT SETUP

In this study, we focus on utilizing LLMs, specifically GPT-4o Mini to summarize counseling transcripts into visual-based summaries. We first developed a web interface that serves as a platform for experimenting prompts using OpenAI's API endpoint. We chose GPT-4o Mini as our primary LLM due to its lightweight nature, cost-effectiveness and strong performance compared to GPT-3.5 Turbo. We designed a prompt that instructs the LLM to summarize the counseling transcript specifically focusing on the patient's responses towards the therapist's questions. Additionally, the prompt includes instructions to format the output according to the PlantUML mind map format. Since the LLM may not be explicitly trained to generate outputs in such a format, we applied a one-shot learning technique by providing a short example of the desired output to guide the model's response. In cases where the output did not meet the desired format, we iteratively refined the prompt by adjusting the instructions and the one-shot example. This method of iteratively experimenting with the prompt has proven to be effective in improving the results as demonstrated by (Zamfirescu-Pereira et al., 2023). After a few iterations of prompt refining, the final devised prompt is used to run the experiment on all our selected samples. Refer to Appendix A.1 for a sample of the prompt. Since the generated output is in PlantUML mind map format which is text-based, we develop a script to convert it into a visual-based interactive mind map using a JavaScript package called jsmind[4].

For this experiment, we randomly sub-sampled 20 samples from the testing set of the MEMO dataset. To validate the feasibility of using LLM for effectively summarizing counseling transcripts and generating outputs in the PlantUML format, we began with a subset of 20 samples. This approach allows us to demonstrate the initial effectiveness of our methodology before scaling to larger datasets.

## 3.3 HUMAN EVALUATION

We created an online evaluation form as part of a human evaluation method to assess the generated visual-based summaries. The evaluation focused on the following 4 criteria:

- *Accuracy*: How well does the visual-based summary reflect the key information from the text-based summary?

- *Completeness*: Does the visual-based summary capture all the important points as presented in the text-based summary?

- *Conciseness*: Is the visual-based summary presented without unnecessary details or redundancy?

- *Coherence*: Is the visual-based summary organized in a logical and understandable way?

These criteria were selected based on evaluation methodologies used in previous studies (So et al., 2024; Van Der Lee et al., 2019; Srivastava et al., 2022; Hochheiser & Verma, 2021; Jin et al., 2024). We invited researchers from the School of Information Technology at Monash University Malaysia to conduct quality evaluation on the 20 generated visual-based summaries. They were asked to assess each generated summary by comparing the visual-based summary with the corresponding text-based summary provided from the dataset using the 4 criteria listed above. The evaluation criteria were presented on a 5-point Likert scale and the survey was designed to provide quantitative feedback only. Additionally, they had the option to view the original transcript as supplementary material.

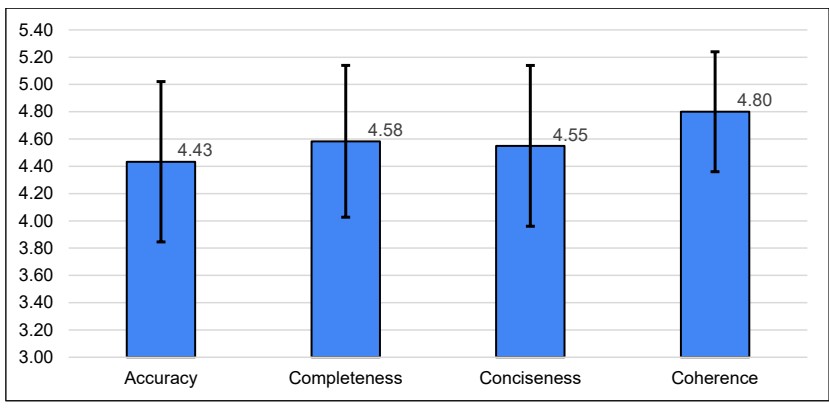

Figure 4: Summary of human evaluation results by criterion.

## 4 RESULTS

A total of 3 participants participated in the survey. Figure 4 demonstrates the overall performance of the visual-based summaries generated from our web-based summarization tool.

**Accuracy**. The accuracy of the visual-based summaries received a mean score of 4.43 ±0.59. This suggests that the participants found the information presented in the mind maps effectively reflects the key points from the original text-based summaries. The standard deviation suggests a high level of agreement among participants, implying that most ratings were clustered around the mean.

**Completeness**. The criterion of completeness received a mean score of 4.58 ±0.56. This indicates that the information presented in the visual-based summaries covers all the necessary details providing a clear overview of the counseling conversation. The low standard deviation indicates that the participants were consistent in their evaluations, reflecting a strong consensus on the model's ability to extract and convey all important details.

**Conciseness**. For conciseness, the model achieved a mean score of 4.55 ±0.59. This score indicates that the visual-based summaries were generally presented without unnecessary details or redundancies. As shown by the standard deviation, evaluations are consistent, suggesting that participants agree that the summaries maintained succinctness in conveying important information.

**Coherence**. The coherence of the visual-based summaries was rated the highest, with a mean score of 4.80 ±0.44. This indicates that the participants found the visual-based summaries to be exceptionally organized and logical. With the organised information, it has facilitated a quick understanding of the content. The low standard deviation proves a strong agreement among the participants, highlighting that the generated mind maps were not only clear but also effectively structured.

Overall, our findings indicate that the web-based summarization tool we developed demonstrates strong potential in summarizing counseling transcripts into visual-based summaries. The high mean scores across all evaluation criteria reflect the effectiveness of the language model in producing visual-based summaries that are accurate, complete, concise, and coherent. However, based on the relatively lower score for accuracy, we observed some cases where the visual-based summaries provided more contextual information about the counseling session than the text-based summaries. This is likely because the visual-based summaries were derived directly from the transcript or dialogues while the text-based summaries were authored in a previous study. This observation highlights a potential advantage of the our generated visual-based summaries which is their ability to incorporate and display additional context from the full transcript. While the text-based summaries published by Srivastava et al. (2022) may prioritize certain aspects of the counseling session such as patient's symptom and history, the visual format allows for a more comprehensive view of the counseling session offering richer contextual insights. Despite the slight variation in accuracy, the overall performance of our web-based summarization tool remains robust, providing a promising indication

---

[4]https://github.com/hizzgdev/jsmind

of the feasibility of using LLMs for generating effective visual-based summaries in mental health counseling contexts.

## 5 DISCUSSION

### 5.1 COMPARISON WITH LEXIMANCER

In this study, we further experiment with the use of Leximancer[5] to analyze counseling session transcripts. Leximancer is a widely used, semi-automated content analysis software designed for qualitative text analysis. It utilizes word occurrence and co-occurrence patterns to map relationships between words, thereby identifying high-level themes from an input text. Leximancer generates concepts and themes by detecting word patterns that frequently appear together within the text (Engstrom et al., 2022). Previous studies in the mental health field such as Fanaian et al. (2013), have employed Leximancer to analyze textual data such as opinions regarding a topic of interest. These generated themes and concepts are then visually presented as an interactive concept map.

We randomly selected one sample from the sub-sample set used in our study and processed it using Leximancer's default settings. By observation and comparison of our generated visual-based summary in Figure 2 with the concept map generated by Leximancer in Figure 5, we found that the themes and concepts presented were somewhat shallow and did not capture the deeper context and underlying meaning of the conversations. This limitation presents a challenge for therapists as they may need to invest additional time in reviewing the results or require specific expertise to effectively use the software for transcript analysis. Our web-based summarization tool addresses this issue by providing a concise summary of the counseling session, offering a more efficient and accessible alternative for therapists.

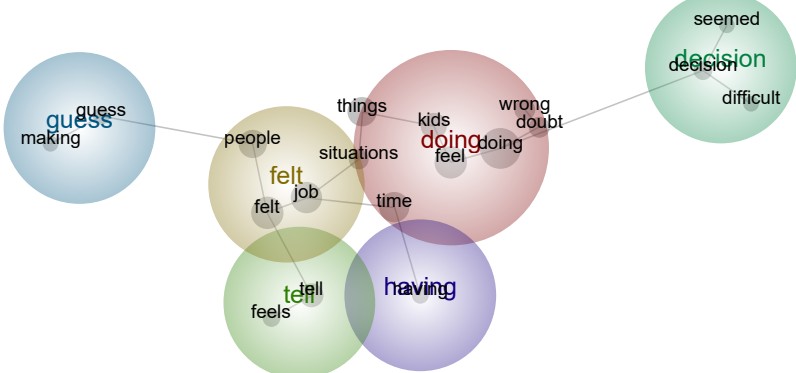

Figure 5: Generated Leximancer concept map where large coloured circles denote the themes and nodes indicate the concepts.

### 5.2 LIMITATIONS

While participants from our human evaluation provided valuable perspectives, the limited number of respondents restricts the generalizability of the results. Additionally, the participants' backgrounds are primarily in technical fields related to LLMs, rather than in mental health. As a result, our study lacks input from professionals or actual users within the mental health sector, which may limit the relevance of the findings to the specific needs and experiences of the mental health community.

## 6 CONCLUSION

In conclusion, we addressed the growing cognitive burden on therapists by introducing a web-based summarization tool leveraging LLMs to generate visual-based summaries from counseling sessions.

---

[5]https://www.leximancer.com/

Based on our human evaluation survey, the results show that our web-based summarization tool can effectively capture and summarize keypoints and format its output suitable for a diagramming tool such as PlantUML. These visual-based summaries provide therapists with a quick overview of counseling sessions aiding in better recall and comprehension. By exploring new possibilities in AI integration, this tool not only enhances therapeutic practice but also holds the potential to contribute to the broader advancement of mental health care. Further development and research such as expanding the user study to include larger sample sizes and participants from diverse mental health roles, including therapists, counselors and psychologists would provide broader insights allowing the tool to offer more tailored support to a wider range of mental health professionals.

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

# A  Appendix

## A.1  Sample prompt used in our study

Based on the counseling transcription below where Therapist is the interviewer and Patient is the interviewee, summarize the conversation, especially Patient's response towards Therapist's questions. Start the output with '@startmindmap' and end with '@endmindmap'. An example is provided. Do not include the content of the example in the final output.

Example Input:
1     Therapist where are you from originally?
2     Patient    atlanta georgia
3     Therapist why'd you move to l_a
4     Patient    um my parents are from here um
5     Therapist what do you do to relax
6     Patient    i like reading books i enjoy i enjoy cooking um exercising is great

Example Output:
@startmindmap
* Conversation Topics
** Patient's Background
*** Patient's origin - Atlanta, Georgia
*** Patient move to L.A.
**** Parents' origin
** Patient's Personality
*** Relaxation activities - Reading books, cooking, exercising
@endmindmap

Input:
1     Therapist So Tommy I wanted to talk to because I was on Facebook the other day and saw some pictures that are kind of concerning to me of you and some of your friends at a party and you know, you're such a good kid, like, I don't know what you're doing drinking. It's not good and it will ruin your life
2     Patient    I mean it was just the like a social gathering it wasn't like I was looking for it the drinks were there and I all my friends were doing it and it was just it's it's not something that I like do okay on occasion but whenever the opportunity presents itself it it's nice to just let loose and have a little bit of fun sometimes
3     Therapist I Tommy there's so many consequences with drinking you know you're not even 21 yet you could get arrested you could you know have a drunk driving accident you could you know, mess up your schoolwork you have such a bright future ahead of you. it just really really concerns me.
4     Patient    I get what you're coming from but I I really did. I thought that I had it under control. And I didn't let myself get too far gone. I didn't want to be in how do I put this little Like, I want it to be in control. And I wanted to make sure that all my friends were Okay, so I didn't let myself drink too much.
5     Therapist There's really no such thing as not drinking too much. I mean, anything that's drinking when you're under age is drinking too much. You know, like, think about it. Think about all the things that can happen to you.
6     Patient    Yeah, I understand exactly what you're saying. And
7     Therapist how much are you drinking?
8     Patient    I don't think I'm drinking that much. I mean, it's, it's mainly for social gatherings. Like it's nothing that I do, like by myself or whatever. It's just the
9     Therapist it's like, every weekend.
10    Patient    every other weekend, I would say,
11    Therapist Tommy, I'm just so concerned, you know, can't you think of anything better to do?
12    Patient    I guess I can, I can probably be more productive in my schoolwork and rather than going out to parties as much and I do get where you're coming from.
13    Therapist Yeah, are your friends the problem? Like maybe you should just start hanging out with more kids in the youth group and not hang out with those kids that you're partying with?
14    Patient    Yeah, I guess I haven't really put it on myself, I guess it's really been forced on their account. And I've just been following in their footsteps just to just because it seems like what they're doing is a lot of fun. And I just want to make sure that I'm getting every bit of the high school experience that I can.
15    Therapist Well, there's a lot of things that seem fun in the moment that we know aren't a good choice, right? That's some of the things we talk about all the time is making good choices for our life and really doing what's best for you. And I just, you know, I really think you should think about making a big change here.
16    Patient    Yeah, I, I do see that. Maybe me drinking not as much would possibly better my life to to a certain degree and I It does make a lot of sense. And I, I guess I can put more effort into sort of lowering the level of drinking.
17    Therapist I think you really need to quit.
18    Patient    Yeah, I mean, alcohol is just so prevalent in our society and it's just around me wherever I go and parties and stuff that I, I kind of feel like I should just get used to it or at least understand its its effects before I should let it get out of my life entirely. I don't I don't really see it as something that will completely destroy my life.
19    Therapist Think you're wrong. You know, it really worries me. Anyway, I wish you would quit. If all you're gonna do is cut down that's better than nothing. You know, I appreciate you talking to me about this, but I'm gonna keep talking to you about it. It's really important. Yeah,
20    Patient    I, I just I don't see the problem in it right now. And maybe it's because I haven't really had that dramatic of an experience. But,

```
21   Therapist um, do you want to wait for that to happen? Do you want to wait for a dramatic experience?
22   Patient    You're right. I don't want I don't want to be too far gone. Before it's too late. Like I, I can imagine something
possibly happening where it's to a degree where I can't escape. And I I guess I've never really had any, any real
experience where alcohol has really affected me that much, or affected anybody close to me that much. So I guess it's
something that I feel like I'm in control of, but in reality, I might not have as much control as I think.
23   Therapist All right. Well, maybe it's something we can work on together. And, you know, I think we should come up
with a plan that's gonna work for you so that you can get to trial here and stop drinking.
24   Patient    Yeah, I think that that would be a good step forward into an A life of possibly being sober and just
experiencing the true the true aspects of life rather than the the haziness of alcohol.
25   Therapist Okay. Sounds good.

Output:
```

Figure 6: Sample prompt used in our study.

### A.2   SAMPLE OUTPUT IN PLANTUML MIND MAP FORMAT

```
@startmindmap
* Conversation Topics
** Patient's Social Behavior
*** Party Attendance
**** Occasional drinking at gatherings
**** Drinking influenced by friends
** Therapist's Concerns
*** Risks of Underage Drinking
**** Legal issues
**** Impact on future and schoolwork
** Patient's Perspective
*** Believes in controlling drinking
**** Drinking mainly at social events
*** Realization of potential problems
**** Understanding need to reduce drinking
** Discussion on Friend Influence
*** Acknowledgment of peer pressure
**** Following friends' actions for fun
** Future Actions
*** Consideration of healthier choices
**** Possible reduction in drinking
**** Emphasizing sobriety for better life experiences
** Collaboration
*** Willingness to work with therapist on a plan
**** Seeking to experience life without alcohol
@endmindmap
```

Figure 7: Generated output in PlantUML mind map format.

## A.3 SAMPLE GENERATED VISUAL-BASED MIND MAP SUMMARY

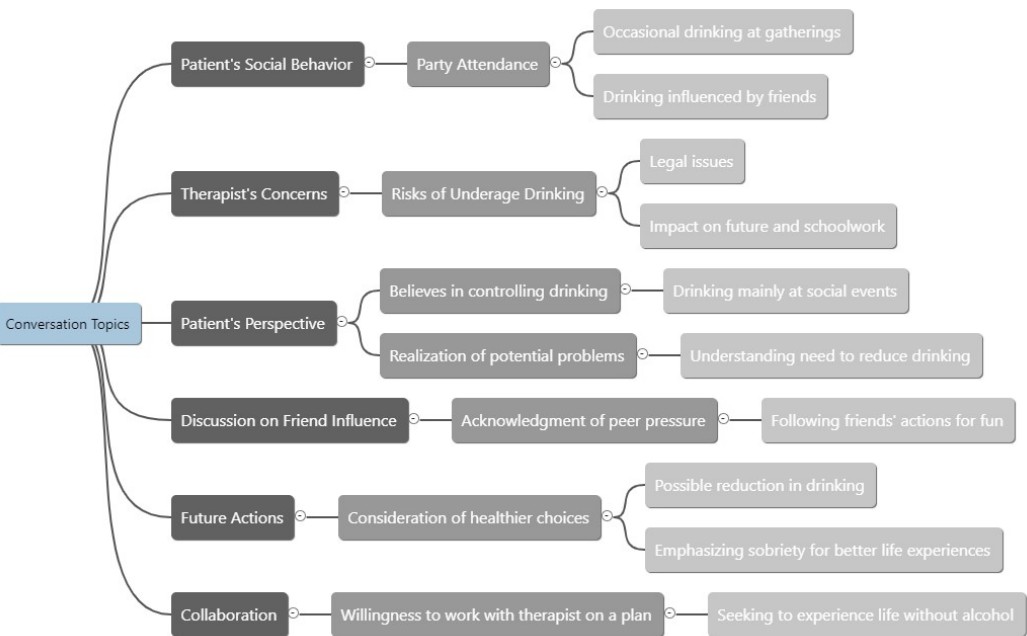

Figure 8: Generated visual-based mind map summary of a counseling session.

