# OpenReview forum: "From Counseling Transcript to Mind Map: Leveraging LLMs for Effective Summarization in Mental Health Counseling"
_ICLR.cc/2025/Conference — Submitted to ICLR 2025_

### Official Review · Reviewer_Jns1 · 2024-11-02

**Soundness:** 2
**Presentation:** 2
**Contribution:** 2
**Rating:** 3
**Confidence:** 4

**Summary:**

This paper presents a tool designed to help mental health therapists by summarizing counseling sessions into mind maps. Using a lightweight language mode GPT-4o minil, the tool takes session transcripts and generates visual summaries that highlight key points in a structured, easy-to-read format. This approach aims to reduce the cognitive load on therapists by providing quick, organized overviews of each session, making it easier to recall important details. The study also includes human evaluations, showing that these visual summaries effectively capture the main elements of a session, potentially enhancing therapeutic practice by simplifying documentation.

**Strengths:**

Originality:
The paper introduces the use of mind maps for summarizing counseling transcripts, which is an interesting shift from typical text summaries. Applying language models to generate these visuals in a mental health context is a practical adaptation, though it builds on existing summarization methods.
Quality:
The method is clearly explained but lacks depth in certain areas. While the pipeline from prompt design to mind map visualization is understandable, the approach could benefit from a stronger methodological foundation or more comprehensive testing. Human evaluations are included, but the evaluation setup could be more rigorous to fully support the tool’s effectiveness.
Clarity:
The paper explains the problem and solution in a straightforward way, with enough detail to understand the main approach. Visual examples help clarify how the mind maps work, although some sections could be streamlined for readability.
Significance:
Addressing the cognitive load for therapists is a relevant issue, and this tool could be helpful in simplifying their workflow. While the impact is practical, the approach is incremental and may inspire further exploration of visual summarization but is unlikely to be transformative on its own.

**Weaknesses:**

1. Limited Sample Size in Evaluation
Although the authors mention that they selected 20 transcripts as a “preliminary” sample, the decision to limit the entire study to these 20 samples is problematic given that the original MEMO dataset contains 212 transcripts. Relying solely on such a small subset, when a much larger set is available, raises concerns about the representativeness of the findings. A larger, more diverse sample could provide a stronger basis for evaluating the tool’s reliability across different counseling scenarios and patient-therapist interactions. Expanding the sample size is essential to enhancing the credibility of the results and giving a fuller picture of the tool's performance.
2. Insufficient Evaluation Criteria and Sample Size of Evaluators
The evaluation approach has two issues: (1) the limited expertise of the evaluators and (2) the small number of evaluators. Referring to the evaluators as “participants” implies they were not specially trained or highly qualified to evaluate counseling summaries, which could compromise the quality of feedback. A robust evaluation for this type of tool typically requires input from domain experts — such as mental health professionals — who can reliably assess criteria like accuracy, relevance, and therapeutic usefulness based on their experience.
Additionally, the study relies on only three evaluators, which is insufficient for a reliable, statistically meaningful assessment. This small evaluator pool, combined with limited experience in mental health, limits the confidence one can have in the evaluation results.
3. Lack of Comparisons with Other Summarization Methods and Models
The paper does not compare mind maps with alternative summarization formats, such as text-based summaries or visual formats like concept maps. This absence weakens the justification for mind maps as the preferred format, as their claimed cognitive benefits remain untested. Additionally, the study only uses GPT-4o Mini, without comparing it to other language models. This limits the scope since other models might perform better or capture nuances differently.

By addressing these issues — expanding the sample size, involving a more qualified evaluator pool, and incorporating comparisons with other methods — the study would achieve a higher standard of rigor and offer stronger evidence for the proposed tool’s effectiveness.

**Questions:**

Why was only a subset of 20 transcripts used for evaluation?
Why were only three “participants” used for evaluation, and what were their qualifications?
Could additional evaluation metrics or objective measures be incorporated?

---

### Official Review · Reviewer_9p3Y · 2024-11-04

**Soundness:** 2
**Presentation:** 3
**Contribution:** 1
**Rating:** 1
**Confidence:** 5

**Summary:**

With the increase in number of patients requiring mental health care, the cognitive load on mental health professionals has increased a lot. A major part of it is referring to notes from the past sessions. Due to visual information being easier to process and remember, the authors propose a visual diagram based approach to summarize past notes instead of just pure text based summaries. They use a simple LLM based prompting mechanism to take in the transcript and output the summary in a structured way that is parsed into a PlantUML mind map syntax. They also build a web tool that can be used for this.

**Strengths:**

- The authors provide a web based tool that can be used by mental health professionals for building mind-map visual diagrams of past notes
- They show that their tool can build reasonably good mind maps evaluated using human experts.

**Weaknesses:**

- The paper lacks novelty - as it is merely an application of using LLMs for a very specific use case. The prompting itself is also not smart or novel in any way as it just has a structured output that is parsed into a mind-map. I dont believe this would be of interest to many people.
- There are no comparisons to text based summaries for this use case and it is not clear how these visual summaries could be more useful to a mental health professional. Several ablations are missing.
- The contributions is really just the prompt and the web based tool.
- There is also no comparison between the use of different kind of models / LLMs use

**Questions:**

- The paper seems like very half-baked at this point and a lot of analysis is missing for this study to be of use to readers.
How do the visual models compare to textual models on information capture ?
Did you ask therapists to review the mind-map models and get feedback ? What was the response like ?

---

### Official Review · Reviewer_6C1b · 2024-11-04

**Soundness:** 2
**Presentation:** 2
**Contribution:** 1
**Rating:** 1
**Confidence:** 4

**Summary:**

The authors propose a method to construct a summarised mind map of counselling transcripts. They use the MEMO dataset, from which the authors extract 20 samples and perform their analysis on them. They developed a web-based tool to showcase the generated mind maps.

While the topic of visual summaries is relevant to explore, this paper tackles it from an engineering perspective rather than a research perspective. This might make a good demo paper in some conference, however, I am not so sure about this being a part of the research main track.

**Strengths:**

- Interesting problem of visual summaries is explored. Mind maps are an effective and quicker way of information communication, so using such things to help therapist is a good idea.
- It is good that the authors have developed a web based tool which can help people.
- The paper is easy to follow.

**Weaknesses:**

- No substantial contribution is there in this paper for it to be a part of main research track, maybe a better fit for demo track for some conference.
- Only 20 mind maps evaluated by only 3 people with using just 1 LLM as the summariser might not be enough to be considered as a thorough investigation. I would suggest the authors to conduct a more detailed analyses by using more LLMs in the place of GPT-4o Mini to summarise the transcripts and then perform a comparison of different LLMs here.
- In a nutshell, the paper does not propose a new problem, does not have a new dataset, does not propose any novel method, and does not contain any interesting observations or analysis. It is a very engineering pov paper, which is not a good fit for this venue.

**Questions:**

- Since the current contribution is low, I think the authors can add another analysis of multilinguality. Maybe they can analyse how different languages affect the mind map creation.

---

### Official Review · Reviewer_cEFZ · 2024-11-09

**Soundness:** 1
**Presentation:** 3
**Contribution:** 2
**Rating:** 3
**Confidence:** 4

**Summary:**

The paper introduces a web-based summarization tool that uses Large Language Models (LLMs), specifically GPT-4o Mini, to transform counseling session transcripts into visual mind map summaries. By leveraging the MEMO dataset of counseling conversations, the tool extracts key points from qualitative dialogues and organizes them hierarchically using PlantUML syntax. Human evaluations based on accuracy, completeness, conciseness, and coherence indicate that the generated mind maps effectively capture essential information. The tool demonstrates potential in aiding therapists to quickly recall session details and reduce cognitive load by providing a clear, visual overview of counseling sessions.

**Strengths:**

The integration of LLMs, specifically GPT-4o Mini, with PlantUML for generating mind maps from qualitative counseling data is a creative and original contribution. This approach moves beyond traditional text-based summaries by providing a visual representation of complex counseling sessions, which is underexplored in the mental health domain.

**Weaknesses:**

1. Limited Sample Size and Representativeness

The study utilizes only 20 randomly selected samples from the MEMO dataset for the evaluation. This small sample size is not sufficiently representative of the diverse range of counseling conversations that occur in real-world settings. As a result, the findings may not generalize well to broader applications. To strengthen the validity of the results, it would be beneficial to include a larger and more varied dataset. This expansion would help in capturing a wider array of counseling scenarios and linguistic nuances, thereby enhancing the robustness of the tool's performance and its applicability to different contexts.

2. Reliance Solely on Human Evaluation Lacking Reproducibility

The evaluation of the generated visual summaries is based entirely on human assessments from a small group of participants who are researchers in the field of Information Technology, not mental health professionals. This approach raises concerns about the reproducibility and objectivity of the results. Human evaluations can be subjective and may vary significantly between different evaluators. Incorporating quantitative evaluation metrics, such as adapted versions of ROUGE or BLEU scores for summarization tasks, could provide more objective measures of the tool's performance. Additionally, involving mental health professionals in the evaluation process would offer insights that are more aligned with practical therapeutic needs, thereby increasing the reliability and validity of the findings.

**Questions:**

1. The authors used only 20 samples in their experiments; have they considered using larger sample sizes to enhance the representativeness of their results? Is future validation of a larger data set planned?

2. In the field of consulting, where there is quite a bit of work on automated summarization [1][2][3], how can it be demonstrated that the presentation of mind maps reduces cognitive load?

[1] Extraction and Summarization of Suicidal Ideation Evidence in Social Media Content Using Large Language Models

[2] Utilizing Large Language Models to Identify Evidence of Suicidality Risk through Analysis of Emotionally Charged Posts

[3] Aligning Large Language Models for Enhancing Psychiatric Interviews through Symptom Delineation and Summarization

---

### Meta-Review · Area_Chair_cF4S · 2024-12-19

**Metareview:**

The paper propose to summarize the counseling transcripts using gpt-4 to construct a Mindmap. The main weaknesses pointed out by reviewers is that the limited sample size of the evaluation dataset (20 transcripts) to meaningfully evaluate the method. The human evaluation should also involve domain experts.

**Additional Comments On Reviewer Discussion:**

NA

---

### Decision · Program_Chairs · 2025-01-22

Reject